# Induced Covariance for Causal Discovery in Linear Sparse Structures

## Abstract

Causal models seek to unravel the cause-effect relationships among variables from observed data, as opposed to mere mappings among them, as traditional regression models do. This paper introduces Sparse Linear Causal Discovery (SLCD), a novel causal discovery algorithm designed for settings in which variables exhibit linearly sparse relationships. In such scenarios, the causal links represented by directed acyclic graphs (DAGs) can be encapsulated in a structural matrix. The proposed approach identifies the correct structural matrix by evaluating how well it reconstructs the data and how closely it satisfies the imposed statistical constraints. This method does not rely on independence tests or graph fitting procedures, making it suitable for scenarios with limited training data. Simulation results on synthetically generated datasets with known linear sparse causal structures show that SLCD consistently outperforms the PC, GES, BIC exact search, and LiNGAM-based methods, achieving average improvements of $35\%$ in precision and $41.5\%$ in recall. Moreover, on the real-world Sachs dataset, SLCD further surpasses these methods in the low-sample-size setting.

## 1 Introduction

Causal learning is an approach used to extract and understand cause-and-effect relationships from data. This approach seeks to uncover the fundamental structures that determine how data are related (Shaska & Mitra, 2025). This structural understanding is at a deeper level than that observed in statistical learning, which is focused on learning various mappings among data (Schölkopf & von Kügelgen, 2022). Discovering causal relations plays a crucial role in the scientific method (Camps-Valls et al., 2023). A comprehensive causal model of a phenomenon could describe the observed data and consistently make predictions. The advantage of this type of learning over statistical learning, which identifies mere associations between variables, lies in its generalization and robustness to distribution changes (Schölkopf et al., 2021). Furthermore, causal relations may be transferable to other problems, which constitutes an additional benefit (Schölkopf et al., 2021).

Several approaches have been proposed for causal discovery (Pearl & Verma, 1995; Spirtes et al., 2001; Shimizu et al., 2006; 2011; Meek, 1997; Chickering, 2002; Yuan & Malone, 2013). These methods are generally classified into two categories (Schölkopf & von Kügelgen, 2022): constraint-based and score-based methods. In constraint-based methods, conditional independencies among variables are tested, and the inferred relations are represented using a DAG that best reflects them. Notable examples of such algorithms include inductive causation (IC) (Pearl & Verma, 1995), Spirtes-Glymour-Scheines (SGS) (Spirtes et al., 2001), Peter-Clark (PC) (Spirtes et al., 2001) and linear non-Gaussian acyclic model (LINGAM) based methods (Shimizu et al., 2006) and (Shimizu et al., 2011). IC and SGS algorithms examine the conditional independencies between each pair of variables conditioned on any subset of the remaining variables and use this information for forming the causal graph. This approach can be computationally intensive due to the large number of subsets. The PC algorithm mitigates this challenge by initiating the search from a complete graph and systematically removing edges, sequentially testing the conditional independencies of each pair and their neighbors. Although the PC algorithm reduces computational cost, it still depends on conditional independence tests, which are computationally demanding and require substantial data, particularly in high-dimensional settings, to produce reliable results. Unlike the previously mentioned methods that rely heavily on conditional independence tests, LiNGAM-based approaches assume non-Gaussianity and linearity in the data. By leveraging these assumptions, LiNGAM aims

to identify the causal graph. Although alleviating the challenge of conditional tests, non-Gaussianity is a limiting assumption.

Alternatively, score-based methods use a scoring function to evaluate graphical representations. Possible graphs are tested against the data, and the graph with the highest score is selected. Some of the prominent methods in this category are greedy equivalent search (GES) (Meek, 1997; Chickering, 2002), and Bayesian information criterion (BIC) exact search (Yuan & Malone, 2013). The primary drawback of these methods (Meek, 1997; Chickering, 2002; Yuan & Malone, 2013) is the exponential growth of the possible graphs as the number of variables (nodes of the graph) increases, which results in higher computation demands.

Another line of work in causal discovery is *causal representation learning* (Schölkopf et al., 2021; Varıcı et al., 2024). In this setting, data are assumed to be generated from high-level latent variables, which are mapped through transformations to the observed data. The objective is to identify both the relations among the latent variables and the transformations that connect them to the observed data. This is carried out in such a way that the resulting representation remains consistent with causal interventions.

However, these approaches do not adequately address scenarios with limited data, where conditional independence tests often fail to yield reliable results. Moreover, causal representation learning typically requires access to both the dataset and its intervened versions, which can be a significant limitation. Therefore, despite the emergence of several methods for causal discovery, there remains no algorithm well-suited for situations involving small datasets and the absence of feasible interventions.

The primary contribution of this paper is the development of a novel causal discovery algorithm designed for linear sparse structures. The key contributions are as follows:

- We propose *Sparse Linear Causal Discovery (SLCD)*, a new algorithm that recovers the structural matrix that encapsulates causal graph information by leveraging induced covariance, data reconstruction, rank, and diagonal structure, specifically for settings where variable relationships can be effectively modeled as sparse linear dependencies.

- We characterize the covariance constraints induced by linear causal dependencies (*induced covariance*). Similar identities have been noted previously in the special case of jointly Gaussian variables (Sullivant et al., 2010); here, we provide a distribution-free formulation.

- We extend the induced covariance framework to accommodate nonlinear causal structures.

- We provide a theoretical analysis of structural matrices that satisfy both induced covariance and reconstruction constraints, establishing results on local uniqueness and sensitivity.

- Through experiments on synthetically generated datasets with known linear sparse causal structures, we show that the proposed method outperforms established causal discovery approaches, achieving on average a $35\%$ improvement in precision and a $41.5\%$ improvement in recall. In addition, experiments on the real-world Sachs dataset (Sachs et al., 2005) demonstrate that the proposed method also surpasses baseline approaches in low-sample-size settings.

The rest of the paper is organized as follows. Section 2 provides a brief overview of the causal learning framework. Section 3 outlines the problem statement and associated challenges, while Section 4 introduces the proposed approach (SLCD). Section 5 discusses the generalization of SLCD to settings with nonlinear causal relations. Section 6 presents the theoretical analysis of structural matrices that satisfy both induced covariance and reconstruction constraints. Section 7 presents and analyzes the simulation results, and Section 8 concludes the paper.

## 2 PRELIMINARIES

To state the problem, we first review the concept of a structural causal model (SCM), a popular method for causal relations modeling (Schölkopf & von Kügelgen, 2022). In this framework, a set of random variables $\mathcal{Y} = \{y_1, y_2, \ldots y_n\}$ is represented as the vertices of a directed acyclic graph (DAG), and the following relations hold:

$$y_i = f_i(\mathcal{P}_i, u_i) \quad \forall i \in \{1, 2, \ldots, n\}, \tag{1}$$

with $f_i$ being a deterministic function, $\mathcal{P}_i \subset \mathcal{Y}$ (parents) represents the set of variables that influence $y_i$, and $u_i$ is exogenous noise, an unobserved external factor (Schölkopf et al., 2021). In the graphical SCM representation, a directed edge exists from each member of $\mathcal{P}_i$ to $y_i$ for $i \in \{1, 2, \ldots, n\}$. The process of causal discovery involves identifying $f_i$ and $\mathcal{P}_i$ for all $n \in \{1, 2, \ldots, n\}$.

# 3 PROBLEM STATEMENT

Let $\boldsymbol{x} = [x_1, x_2, \ldots, x_n]^T \in \mathbb{R}^n$ be a vector that contains all the random variables for which causal relationships are to be discovered. These variables fall into two categories: independent (causal) variables, each of which has no parents, and dependent (effect) variables, which are influenced by the causal variables through the SCM. We define $\mathcal{D}$ as the set of indices for dependent variables. According to the SCM, each $x_i$ for $i \in \mathcal{D}$ is a function of a subset of independent variables $\mathcal{P}_i$, which are considered the parents of $x_i$. We assume that the functions $f_i$ for all $i \in \{1, 2, \ldots, n\}$ are linear and that $|\mathcal{P}_i| \leq \tau$, where $|\cdot|$ denotes set cardinality and $\tau$ is a model parameter. We also assume that no exogenous noise is present; that is, the values of the dependent variables are fully determined once the values of their parent variables are known, a situation that arises naturally in many problems across domains such as physics and biology (Li et al., 2024; Yang et al., 2022).

Under these assumptions, we can represent $x_i$ as follows:

$$x_i = \boldsymbol{d}_i^T \boldsymbol{x}, \tag{2}$$

where $\boldsymbol{d}_i \in \mathbb{R}^n$ is a vector with no more than $\tau$ non-zero elements, corresponding to the independent variables upon which $x_i$ depends. Given this notation, the relationship for all variables can be expressed as

$$\boldsymbol{x} = \boldsymbol{D}\boldsymbol{x}, \tag{3}$$

where $\boldsymbol{D} \in \mathbb{R}^{n \times n}$ is a matrix constructed from the vectors $\boldsymbol{d}_i, \forall i \in \{1, 2, \ldots, n\}$ as its rows. The matrix $\boldsymbol{D}$ contains all pertinent information on the causal structure of this model, and an accurate estimation of $\boldsymbol{D}$ reveals the underlying causal relations.

In practice, there is often limited or no prior knowledge about the underlying structure of the data, and only the dataset itself is available. We use $\boldsymbol{X} = [\boldsymbol{x}_1, \boldsymbol{x}_2, \ldots \boldsymbol{x}_m] \in \mathbb{R}^{n \times m}$ to represent the given dataset, where each $\boldsymbol{x}_i \in \mathbb{R}^n$ is a sample. Applying (3), we have

$$\boldsymbol{X} = \boldsymbol{D}\boldsymbol{X}. \tag{4}$$

The primary objective is to determine the causal structure ($\boldsymbol{D}$) from $\boldsymbol{X}$. Our goal is to develop a procedure to estimate $\boldsymbol{D}$ without using conditional independence tests, making it suitable when the number of data samples are limited, especially in high-dimensional data.

## 3.1 CHALLENGES

In the estimation of $\boldsymbol{D}$, several challenges must be addressed. Based on the previous discussion, it can be inferred that $\boldsymbol{D}$ must satisfy the condition expressed in (3). However, as demonstrated in the following example, this condition alone is insufficient for uniquely determining the causal structure.

**Example 1.** Suppose data is created as follows

$$\begin{bmatrix} x_1 \\ x_2 \\ x_3 \end{bmatrix} = \begin{bmatrix} 1 & 0 & 0 \\ 0 & 1 & 0 \\ 1 & 1 & 0 \end{bmatrix} \begin{bmatrix} x_1 \\ x_2 \\ x_3 \end{bmatrix}. \tag{5}$$

This structure suggests that $x_1$ and $x_2$ are independent variables (as they are not linear combinations of any other variables), while $x_3$ is the sum of $x_1$ and $x_2$. When only the data is available and the objective is to satisfy the condition given in (3), the solution may not be unique. For example, the identity matrix ($\boldsymbol{I} \in \mathbb{R}^{3 \times 3}$) and the following matrix also satisfies (3) for the aforementioned setup:

$$\begin{bmatrix} 0 & -1 & 1 \\ 0 & 1 & 0 \\ 0 & 0 & 1 \end{bmatrix}. \tag{6}$$

This example illustrates that observational data alone is insufficient to uniquely determine the causal relations. In general, when only observational data are available, and no additional assumptions

are imposed, multiple causal graphs can generate the same data (Spirtes et al., 2001). Furthermore, this example shows how causal discovery differs from a regression problem. While both solutions may be acceptable in the context of regression, only one solution reveals the underlying causal structure. In other words, what separates this approach from a linear algebra regression is that we are not looking for any solution of (3) but the one that describes the associations according to the underlying causal structural matrix.

## 4 INDUCED COVARIANCE-BASED CAUSAL DISCOVERY

To address the challenges discussed above, it is beneficial to explore certain properties the causal structural matrix $D$. The structure of $D$ can provide valuable insights into the relations among variables. Any $k^{\text{th}}$ row of the structural matrix $D$, whose elements are all equal to zero except for the element in column $k$, which is equal to 1, corresponds to an independent random variable. Since all variables are linear combinations of the independent variables, the rank of $D$ is equal to the number of independent variables. This observation suggests that the structure of $D$ can be used for narrowing down the number of potential solutions for $D$.

To further constrain the possible solutions for $D$, the following theorem establishes a connection between the statistical properties of the data and the structural matrix (Sullivant et al., 2010). More specifically, it shows that selecting a specific value for the variable $D$, uniquely determines the value of the covariance matrix of the data, indicating that $D$ imposes a constraint on the covariance matrix.

**Theorem 1.** Let $D \in \mathbb{R}^{n \times n}$ denote the causal structural matrix that describes the dependencies among the cause and effect variables stacked in the zero-mean random vector $x = [x_1, x_2, \ldots, x_n]^T \in \mathbb{R}^n$, that is $x = Dx$. Then the covariance matrix of $x$ is given by $D\sigma D^T$, in which $\sigma \in \mathbb{R}^{n \times n}$ is diagonal, with diagonal element $\sigma_{ii} = \text{Var}(x_i)$ for every $i \in \{1, 2, \ldots, n\}$.

*Proof.* Since the components of $x$ have zero mean, we have $\text{Cov}(x_i, x_j) = \mathbb{E}[x_i x_j]$, for all $i, j \in \{1, 2, \ldots, n\}$. Under the assumed linear structure, we have

$$x_i = d_i^T x, \qquad x_j = d_j^T x,$$

where $d_i^T$ and $d_j^T$ denote the $i$-th and $j$-th rows of $D$, respectively. Then we have

$$\text{Cov}(x_i, x_j) = \mathbb{E}[x_i x_j] = \mathbb{E}[d_i^T x d_j^T x]. \tag{7}$$

Since the only nonzero elements in $d_i^T x d_j^T x$ occurs when both $d_j$ and $d_i$ have non-zero elements in the same positions, i.e., sharing the same independent (causal) variable, we have

$$\mathbb{E}[x_i, x_j] = d_i^T \sigma d_j. \tag{8}$$

By applying the same procedure to all $(i, j)$ pairs, the theorem is proven. $\square$

This theorem restricts the solutions of (3) by imposing that the correct solution must not only satisfy (3) but also fulfill the condition $\Sigma = D\sigma D^T$, where $D\sigma D^T$ is the induced covariance by $D$ and $\Sigma \in \mathbb{R}^{n \times n}$ is the covariance matrix of data, which can be estimated directly from data.

By using the properties of $D$ and its implications on the structure of data, we can formulate the following optimization problem for structure recovery:

$$\arg \min_D \{\text{rank}(D) + \lambda \text{Tr}(D)\}$$
$$\text{subject to} \quad X = DX,$$
$$\Sigma = D\sigma D^T, \tag{9}$$
$$\|d_i^T\|_0 \leq \tau \quad \forall i \in \{1, 2, \ldots, n\}.$$

In this formulation, $D \in \mathbb{R}^{n \times n}$ represents the structural matrix, while $X = [x_1, x_2, \ldots x_m] \in \mathbb{R}^{n \times m}$ is the dataset. The covariance matrix of the data is represented by $\Sigma \in \mathbb{R}^{n \times n}$, and $\sigma \in \mathbb{R}^{n \times n}$ is a diagonal matrix whose diagonal entries correspond to those of $\Sigma$. The term $d_i^T$ represents the $i$-th row of $D$. The operator $\|\cdot\|_0$ returns the number of non-zero elements in a vector. Additionally, $\tau$ controls the number of independent variables, and $\lambda$ serves as a scaling parameter. The rank$(D)$

term prevents the model from becoming overly complicated, and $\text{tr}(\boldsymbol{D})$ discourages the solution from being close to the $\boldsymbol{I}$, which implies all variables are independent.

Rank (the number of non-zero singular values) requires combinatorial calculation, which makes the problem intractable. To address this challenge, the idea proposed in (Mohimani et al., 2009) is used, which approximates $\|.\|_0$ as:

$$\|x\|_0 \approx 1 - e^{-\frac{x^2}{\sigma^2}}. \tag{10}$$

By combining these ideas, the final problem formulation is

$$\arg\min_{D}\{\sum_{i=1}^{n}(1 - e^{-\frac{s_i^2}{\sigma^2}}) + \lambda \sum_{i=1}^{n}(1 - e^{-\frac{d_{(i,i)}^2}{\sigma^2}})\}$$
$$\text{subject to} \quad \boldsymbol{X} = \boldsymbol{D}\boldsymbol{X}, \tag{11}$$
$$\boldsymbol{\Sigma} = \boldsymbol{D}\boldsymbol{\sigma}\boldsymbol{D}^T,$$
$$\|\boldsymbol{d}_i^T\|_0 \leq \tau \quad \forall i \in \{1, 2, \ldots, n\},$$

where $s_i, \forall i \in \{1, 2, \ldots, n\}$ are the singular values of $\boldsymbol{D}$.

To present the final algorithm for obtaining the solution of (11), it is necessary to consider $\|\boldsymbol{d}_i^T\|_0 \leq \tau \quad \forall i \in \{1, \ldots, n\}$, which also requires combinatorial calculations. To handle that, we propose solving the following optimization problem:

$$\arg\min_{D}\{\sum_{i=1}^{n}(1 - e^{-\frac{s_i^2}{\sigma^2}}) + \lambda \sum_{i=1}^{n}(1 - e^{-\frac{d_{(i,i)}^2}{\sigma^2}})\}$$
$$\text{subject to} \quad \boldsymbol{X} = \boldsymbol{D}\boldsymbol{X}, \tag{12}$$
$$\boldsymbol{\Sigma} = \boldsymbol{D}\boldsymbol{\sigma}\boldsymbol{D}^T,$$

and for each row of the resulting $\boldsymbol{D}$, we retain only the $\tau$ entries with the largest absolute values. This process is iterated $N$ times.

Due to noise effects on data, (12) might not have a solution, therefore, some relaxation on the constraint might be required. This is done as follows:

$$\arg\min_{D}\{\sum_{i=1}^{n}(1 - e^{-\frac{s_i^2}{\sigma^2}}) + \lambda \sum_{i=1}^{n}(1 - e^{-\frac{d_{(i,i)}^2}{\sigma^2}})\}$$
$$\text{subject to} \quad \|\boldsymbol{X} - \boldsymbol{D}\boldsymbol{X}\|_F^2 \leq \epsilon_1, \tag{13}$$
$$\|\boldsymbol{\Sigma} - \boldsymbol{D}\boldsymbol{\sigma}\boldsymbol{D}^T\|_F^2 \leq \epsilon_2,$$

where $\epsilon_1$ and $\epsilon_2$ can be tuned to achieve the best result.

Solving (11) requires an initial estimate for $\boldsymbol{D}$, and the final value of the objective function depends on this initial estimate. To obtain the optimal solution, we propose executing the algorithm multiple times, each with a distinct random initialization. The solution that yields the lowest value of the objective function is then retained as the final result. The SLCD algorithm pseudocode is presented in appendix A.

## 5 SLCD, BEYOND LINEARITY

The proposed framework can be further extended to scenarios in which the SCMs governing the causal relations are nonlinear. The idea relies on the Taylor series expansion of the governing function. Suppose $x_i = h(\boldsymbol{x})$, where $x_i$ is the $i^{\text{th}}$ entry of $\boldsymbol{x}$ and is causally related to $\boldsymbol{x}$ through the deterministic function $h : \mathbb{R}^n \to \mathbb{R}$. Assuming $h \in C^{\infty}(\mathbb{R}^n)$ (the set of infinitely differentiable functions on $\mathbb{R}^n$), the Taylor series expansion implies that $x_i$ can be expressed as a polynomial in the entries of $\boldsymbol{x}$. Based on this observation, the following theorem establishes the induced covariance for this scenario.

**Theorem 2.** Suppose $\boldsymbol{x} = g(\boldsymbol{x})$, where $\boldsymbol{x} \in \mathbb{R}^n$ is a vector of random variables and $g : \mathbb{R}^n \to \mathbb{R}^n$ represents the causal relations such that $g_i \in C^\infty(\mathbb{R}^n)$ for all $i \in \{1, 2, \ldots, n\}$. Then, $\boldsymbol{x}$ can be represented as $\boldsymbol{x} = \sum_{i=1}^\infty \boldsymbol{D}_i \boldsymbol{x}^i$, where $\boldsymbol{D}_i \in \mathbb{R}^{n \times n}$ denotes the coefficient matrices for all $i \in \{1, 2, \ldots, n\}$, and $\boldsymbol{x}^i = (x_1^i, \ldots, x_n^i)^T$ is the vector obtained by raising each entry of $\boldsymbol{x}$ to the $i$-th power. The covariance matrix of $\boldsymbol{x}$ is then given by

$$\boldsymbol{\Sigma} = \sum_{i=1}^\infty \sum_{i=j}^\infty \boldsymbol{D}_i \boldsymbol{\sigma}_{ij} \boldsymbol{D}_j^T, \tag{14}$$

where $\boldsymbol{\sigma}_{ij}$ is a diagonal matrix with its $l^{\text{th}}$ diagonal elements be $\mathbb{E}[x_l^i x_l^j]$ for all $l \in \{1, 2, \ldots, n\}$.

The proof is deferred to Appendix C.1. Similar to the linear case discussed previously, Theorem 2 provides a way to formulate an optimization problem with a constraint stronger than simply minimizing the reconstruction error, i.e., regression. To formulate the optimization problem for the nonlinear case, let $\boldsymbol{X} \in \mathbb{R}^{n \times m}$ be the dataset containing $m$ samples. One can then determine the coefficient matrices that satisfy the following equations:

$$\boldsymbol{X} = \sum_{i=1}^\infty \boldsymbol{D}_i \boldsymbol{X}^i, \tag{15}$$

$$\boldsymbol{\Sigma} = \sum_{i=1}^\infty \sum_{i=j}^\infty \boldsymbol{D}_i \boldsymbol{\sigma}_{ij} \boldsymbol{D}_j^T, \tag{16}$$

where $\boldsymbol{X}^i$ denotes the elementwise $i$-th power of $\boldsymbol{X}$, $\boldsymbol{\Sigma}$ is the covariance matrix of the data, and $\boldsymbol{\sigma}_{ij}$ is a diagonal matrix whose $l^{\text{th}}$ diagonal entry is given by $\mathbb{E}[x_l^i x_l^j]$ for all $l \in \{1, 2, \ldots, n\}$.

# 6 INDUCED-COVARIANCE: THEORETICAL ANALYSIS

In this section, a theoretical analysis of the proposed method is presented by examining the behavior of solutions that satisfy both the reconstruction and induced covariance constraints. The analysis focuses on the local properties of these solutions, in particular on whether they correspond to isolated points or manifolds, as well as on their sensitivity to the perturbations in the covariance matrix.

**Theorem 3** (Local Uniqueness of Solutions). Let $F(\boldsymbol{D}) = \boldsymbol{\Sigma} - \boldsymbol{D}\boldsymbol{\sigma}\boldsymbol{D}^T$, where $\boldsymbol{\Sigma} \in \mathbb{R}^{n \times n}$ is the covariance matrix of the data and $\boldsymbol{\sigma} \in \mathbb{R}^{n \times n}$ is its diagonal part. Let $\boldsymbol{X} \in \mathbb{R}^{n \times m}$ be the data matrix, and define $\mathcal{S} = \{\boldsymbol{\Delta} \in \mathbb{R}^{n \times n} : \boldsymbol{\Delta}\boldsymbol{X} = 0\}$. Assume that $\boldsymbol{D}^*$ satisfies $F(\boldsymbol{D}^*) = \boldsymbol{0}$ and $\boldsymbol{D}^*\boldsymbol{X} = \boldsymbol{X}$, and suppose that no nonzero $\boldsymbol{\Delta} \in \mathcal{S}$ makes $\boldsymbol{D}^*\boldsymbol{\sigma}\boldsymbol{\Delta}^T$ skew-symmetric. Then there exists $r > 0$ such that any $\boldsymbol{D}$ with $F(\boldsymbol{D}) = \boldsymbol{0}$, $\boldsymbol{D}\boldsymbol{X} = \boldsymbol{X}$, and $\|\boldsymbol{D} - \boldsymbol{D}^*\|_F < r$ must satisfy $\boldsymbol{D} = \boldsymbol{D}^*$.

The proof is provided in Appendix C.2. This theorem establishes that, under suitable conditions on $\boldsymbol{D}^*\boldsymbol{\sigma}\boldsymbol{\Delta}^T$, the solutions of $F(\boldsymbol{D}) = \boldsymbol{0}$ subject to $\boldsymbol{D}\boldsymbol{X} = \boldsymbol{X}$ is locally unique. Next, the following theorem analyzes the sensitivity of the solutions of $F(\boldsymbol{D}) = \boldsymbol{0}$ to perturbations of the covariance matrix.

**Theorem 4** (Sensitivity Under Covariance Perturbations). Let $F(\boldsymbol{D}) = \boldsymbol{\Sigma} - \boldsymbol{D}\boldsymbol{\sigma}\boldsymbol{D}^\top$, where $\boldsymbol{\Sigma} \in \mathbb{R}^{n \times n}$ is the covariance matrix of data and $\boldsymbol{\sigma} \in \mathbb{R}^{n \times n}$ is its diagonal part. Suppose $\boldsymbol{D}^*$ satisfies $F(\boldsymbol{D}^*) = \boldsymbol{0}$ and $\boldsymbol{D}^*\boldsymbol{X} = \boldsymbol{X}$, where $\boldsymbol{X} \in \mathbb{R}^{n \times m}$ is the data matrix. Define $\mathcal{S} = \{\boldsymbol{\Delta} \in \mathbb{R}^{n \times n} \mid \boldsymbol{\Delta}\boldsymbol{X} = \boldsymbol{0}\}$. Assume that no nonzero $\boldsymbol{\Delta} \in \mathcal{S}$ makes $\boldsymbol{D}^*\boldsymbol{\sigma}\boldsymbol{\Delta}^\top$ skew-symmetric. Let $c$ denote the smallest singular value of $\boldsymbol{D}^*\boldsymbol{\sigma}\boldsymbol{\Delta}^\top + \boldsymbol{\Delta}\boldsymbol{\sigma}\boldsymbol{D}^{*\top}$ restricted to $\mathcal{S}$. Consider now a perturbed covariance matrix $\boldsymbol{\Sigma} + \Delta\boldsymbol{\Sigma}$, and let $\boldsymbol{D}$ satisfy the perturbed equation

$$\boldsymbol{\Sigma} + \Delta\boldsymbol{\Sigma} - \boldsymbol{D}\boldsymbol{\sigma}\boldsymbol{D}^\top = \boldsymbol{0}.$$

If $\|\Delta\boldsymbol{\Sigma}\|_F < \frac{c^2}{4\|\boldsymbol{\sigma}\|_2}$, then $\boldsymbol{D}$ cannot satisfy

$$\frac{c - \sqrt{c^2 - 4\|\boldsymbol{\sigma}\|_2 \|\Delta\boldsymbol{\Sigma}\|_F}}{2\|\boldsymbol{\sigma}\|_2} \;<\; \|\boldsymbol{D} - \boldsymbol{D}^*\|_F \;<\; \frac{c + \sqrt{c^2 - 4\|\boldsymbol{\sigma}\|_2 \|\Delta\boldsymbol{\Sigma}\|_F}}{2\|\boldsymbol{\sigma}\|_2}.$$

The proof is provided in Appendix C.3. This theorem establishes a non-feasible region for the distance between the perturbed and unperturbed solutions of the induced covariance equation.

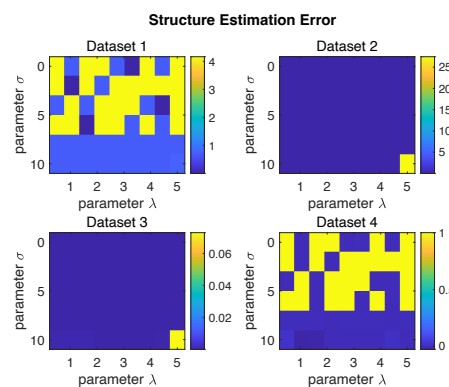

Figure 1: Structure estimation error for various datasets and various hyperparameters.

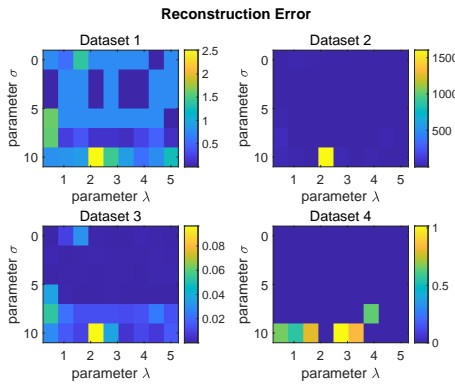

Figure 2: Reconstruction error for various datasets and various hyperparameters.

If there exist values of $\mathbf{\Delta} \in \mathcal{S} \setminus \{\mathbf{0}\}$ such that $\boldsymbol{D}^* \boldsymbol{\sigma} \boldsymbol{\Delta}^T$ is skew-symmetric, then the nearby solutions may not be isolated; that is, there may exist manifolds of solutions along those directions. This necessitates additional restrictions on the solutions to distinguish these directions, making the use of regularization essential in such cases. For this reason, SLCD penalizes the solutions based on their rank and trace in order to further enforce constraints that yield isolated solutions.

## 7 SIMULATION RESULTS AND ANALYSIS

This section presents the results of our simulation studies, conducted on both synthetic and real-world datasets. The synthetic datasets are generated from a known causal structure matrix, and the evaluation examines how accurately the proposed method recovers this underlying structure. For real-world validation, we use the Sachs dataset (Sachs et al., 2005) and assess the ability of the method to identify the corresponding causal pathways. For benchmarking, we evaluate our approach against the PC, GES, LiNGAM-ICA, LiNGAM-Direct, and BIC exact search causal discovery methods.

For comprehensive reporting, we evaluate following metrics: the data reconstruction error, the recovery error of the causal matrix, the recovery error of the covariance matrix, precision (proportion of the number of correct estimated links to the total number of estimated links), and recall (proportion of the number of correct estimated links to the total number of links in the true graph). Let $\hat{\boldsymbol{D}} \in \mathbb{R}^{n \times n}$ be the estimated matrix obtained from the proposed algorithm, and $\boldsymbol{X} \in \mathbb{R}^{n \times m}$ represent the training data. The reconstruction error is then defined as:

| Dataset | IV count[*] | $x_1$ | $x_2$ | $x_3$ | Other Entries |
|---------|-------------|-------|-------|-------|---------------|
| Dataset 1 | 1 | U(-2.5, 2.5) | $2x_1$ | $0.4x_1$ | - |
| Dataset 2 | 2 | U(-2.5, 2.5) | U(-2.5, 2.5) | $0.3x_1$ | $[x_4] = \begin{bmatrix} 1 & 2 \end{bmatrix} \begin{bmatrix} x_1 \\ x_2 \end{bmatrix}$ |
| Dataset 3 | 2 | U(-2.5, 2.5) | U(-2.5, 2.5) | $x_1 + 3x_2$ | $\begin{bmatrix} x_4 \\ x_5 \end{bmatrix} = \begin{bmatrix} 2 & 3 & 0 \\ 0 & 2 & 0.5 \end{bmatrix} \begin{bmatrix} x_1 \\ x_2 \\ x_3 \end{bmatrix}$ |
| Dataset 4 | 3 | U(-2.5, 2.5) | U(-2.5, 2.5) | N(0,4) | $\begin{bmatrix} x_4 \\ x_5 \\ x_6 \end{bmatrix} = \begin{bmatrix} 1 & 0 & 0.3 \\ 2 & 3 & 0 \\ 0 & 2 & 0.5 \end{bmatrix} \begin{bmatrix} x_1 \\ x_2 \\ x_3 \end{bmatrix}$ |
| Dataset 5 | 3 | U(-2.5, 2.5) | U(-2.5, 2.5) | N(0,4) | $\begin{bmatrix} x_4 \\ x_5 \\ x_6 \\ x_7 \end{bmatrix} = \begin{bmatrix} 1 & 0 & 0.5 \\ 0 & 1 & 2 \\ 1 & 0 & 3 \\ 0 & 1 & 1 \end{bmatrix} \begin{bmatrix} x_1 \\ x_2 \\ x_3 \end{bmatrix}$ |

Table 1: Dataset Information. [*] IV count refers to the number of independent variables in each dataset.

| Method | Precision (%) | Recall (%) | $F_1$ (%) |
|--------|---------------|------------|-----------|
| PC | 44 | 24 | 31 |
| GES | 36 | 29 | 32 |
| LiNGAM-ICA | 29 | 41 | 34 |
| LiNGAM-Direct | 29 | 47 | 36 |
| SLCD | 42 | 47 | 44 |

Table 2: Performance comparison on the Sachs dataset when only $10\%$ of the samples are used for each algorithm. SLCD parameters: $\lambda = 5 \times 10^{-5}$, $\sigma = 0.01$, $\tau = 2$.

$$\frac{1}{nm}\|\boldsymbol{X} - \hat{\boldsymbol{D}}\boldsymbol{X}\|_F^2. \tag{17}$$

We define the true structural matrix as $\boldsymbol{D} \in \mathbb{R}^{n \times n}$ and, thus, the recovery error of structural matrix will be:

$$\frac{1}{n^2}\|\boldsymbol{D} - \hat{\boldsymbol{D}}\|_F. \tag{18}$$

Let $\boldsymbol{\Sigma} \in \mathbb{R}^{n \times n}$ represent the true covariance matrix of the original data with $\boldsymbol{\Sigma} \in \mathbb{R}^{n \times n}$, then the recovery error of the covariance matrix is defined as

$$\frac{1}{n^2}\|\boldsymbol{\Sigma} - \hat{\boldsymbol{D}}\sigma\hat{\boldsymbol{D}}^T\|_F, \tag{19}$$

in which, $\boldsymbol{\sigma}$ is a diagonal matrix with diagonal elements of $\boldsymbol{\Sigma}$.

## 7.1 SYNTHETIC DATA

For simulation on synthetic data, five distinct datasets were generated, henceforth referred to as Dataset 1, Dataset 2, Dataset 3, Dataset 4, and Dataset 5. Each dataset comprises 1000 samples. Table 1 provides detailed information on the generation process for each dataset. The variables $x_i$, where $i \in \{1, 2, \ldots, 7\}$, represent the elements of the data vector $\boldsymbol{x} = [x_1, x_2, \ldots, x_7]^T$. The presence of a '-' symbol in place of a variable indicates its absence from the corresponding dataset, reflecting the varying dimensionality across datasets. The table indicates the data distribution from which the samples of independent variables are drawn. For dependent variables, the table specifies the linear combinations used to generate them.

$U(a, b)$ represents the uniform distribution of data in the $[a, b]$ interval. $N(\mu, \sigma^2)$ represents a Gaussian random variable with mean $\mu$ and variance $\sigma^2$. By constructing the datasets in this manner, the variables exhibit the linear sparse relations that SLCD is specifically designed to handle. This



Figure 3: Covariance matrix estimation error for various datasets and various hyperparameters.

| Method | Precision (%) | Recall (%) | $\mathbf{F_1}$ (%) |
|---|---|---|---|
| PC | 55 | 35 | 43 |
| GES | 20 | 24 | 22 |
| LiNGAM-ICA | 21 | 53 | 30 |
| LiNGAM-Direct | 31 | 65 | 42 |
| SLCD | 32 | 53 | 40 |

Table 3: Performance comparison on the Sachs dataset when all of the samples are used for each algorithm. SLCD parameters: $\lambda = 0.01$, $\sigma = 1$, $\tau = 2$.

approach also enables the evaluation of algorithm performance across various data dimensions. Additionally, the datasets include independent variables with different data distributions, allowing for the assessment of algorithm robustness under diverse distributional scenarios. It is important to note that for the dependent variables, each linear combination results in a convolution of the data distributions, further contributing to the variability in the distributions of the dataset's variables.

Figures 1 through 3 display the algorithm's simulation results on performance metrics for various hyperparameter settings. The results reveal moderate sensitivity to hyperparameters, with effective recovery of the underlying structure when parameters are chosen appropriately. The figures also indicate a broad range of satisfactory parameters, demonstrating the method's robustness. The detailed performance of SLCD in recovering the structural matrix of each dataset for the hyperparameter pair $(\sigma, \lambda) = (0.3, 5)$ is presented in Table 5 in the Appendix B. It shows that the method successfully recovers the structural matrix of all datasets, with the exception of Dataset 1.

Table 4 presents the simulation results of SLCD in comparison with several well-known causal discovery algorithms. The results indicate that SLCD outperforms the other methods by an average of $35\%$ in precision and $41.5\%$ in recall across Datasets 2 through 5. While all methods exhibit challenges with Dataset 1, SLCD consistently demonstrates superior performance in the remaining datasets.

SLCD demonstrates suboptimal performance on Dataset 1. This can be attributed to the structure of Dataset 1, wherein only one independent variable exists, and all other variables are scalar multiples thereof. This configuration does not provide sufficient information to unambiguously identify the independent variable, as any of the variables could potentially fulfill this role. This ambiguity introduces uncertainty into the algorithm, potentially leading to diverse solutions. However, as the structural complexity increases with the introduction of additional independent variables, the informational content of the data becomes more robust, facilitating more accurate recovery of the underlying causal structure.

| Method | PC | GES | LG ICA | LG Direct | BIC Search | SLCD |
|---|---|---|---|---|---|---|
| Precision (%) | 33 | 50 | 0 | 33 | 0 | 0 |
| Recall (%) | 100 | 50 | 0 | 50 | 0 | 0 |
| Number of Correct link estimation | 2 | 1 | 0 | 1 | 0 | 0 |
| Dataset 1 | | | | | | |
| Method | PC | GES | LG ICA | LG Direct | BIC Search | SLCD |
| Precision (%) | 50 | 60 | 25 | 0 | 75 | 100 |
| Recall (%) | 66 | 100 | 33 | 0 | 100 | 100 |
| Number of Correct link estimation | 2 | 3 | 1 | 0 | 3 | 3 |
| Dataset 2 | | | | | | |
| Method | PC | GES | LG ICA | LG Direct | BIC Search | SLCD |
| Precision (%) | 37 | 43 | 0 | 0 | 43 | 100 |
| Recall (%) | 60 | 60 | 0 | 0 | 60 | 100 |
| Number of Correct link estimation | 3 | 3 | 0 | 0 | 3 | 5 |
| Dataset 3 | | | | | | |
| Method | PC | GES | LG ICA | LG Direct | BIC Search | SLCD |
| Precision (%) | 100 | 100 | 20 | 10 | 67 | 100 |
| Recall (%) | 100 | 100 | 33 | 17 | 100 | 100 |
| Number of Correct link estimation | 6 | 6 | 2 | 1 | 6 | 6 |
| Dataset 4 | | | | | | |
| Method | PC | GES | LG ICA | LG Direct | BIC Search | SLCD |
| Precision (%) | 30 | 75 | 8 | 13 | 54 | 100 |
| Recall (%) | 37 | 75 | 12 | 25 | 100 | 100 |
| Number of Correct link estimation | 3 | 6 | 1 | 2 | 6 | 8 |
| Dataset 5 | | | | | | |

Table 4: Performance comparison of PC, GES, LG ICA (LINGAM IC), LG Direct (LINGAM Direct), BIC Exact Search, and SLCD algorithms.

## 7.2 REAL-WORLD DATASET

Tables 2 and 3 present the performance of the competing methods on the Sachs dataset under different sample-size conditions, evaluated in terms of precision, recall, and $F_1$ score (the harmonic mean of precision and recall). These experiments are designed to assess how each method performs when different amounts of data are available.

Table 2 reports the results when only 10% of the dataset is provided to each algorithm. In this setting, the sample subset is selected uniformly at random from the full dataset and used as the input for each method. As the results indicate, SLCD achieves the highest performance under this limited-data regime, largely because it does not rely on conditional independence tests, which typically require larger sample sizes to produce reliable outcomes.

Table 3 shows the results obtained when the full dataset is used. In this case, SLCD performs competitively with the baseline methods. Although SLCD is designed for linear sparse structures, the results suggest that it retains a degree of adaptability even when the underlying system, such as the Sachs dataset, does not fully conform to a linear model.

## 8 CONCLUSION

This paper proposes an algorithm for causal discovery within a linear sparse structure, leveraging properties of the causal structure matrix, specifically its rank, which reflects the number of independent variables, and the notion of induced covariance. Simulation studies confirm the algorithm's effectiveness across diverse configurations. Our next direction is to refine the independence criteria. While induced covariance was useful, it does not fully ensure independence across all the scenarios. Addition of stronger constraints can further enhance this framework. This extension would enhance the algorithm's versatility and reliability across a wider range of applications and data types.

THE USE OF LARGE LANGUAGE MODELS (LLMS) STATEMENT

Large language models were used exclusively for language refinement, including proofreading and grammatical correction.

REPRODUCIBILITY STATEMENT

To ensure reproducibility, we provide the pseudocode of the proposed algorithm in Appendix A. The complete implementation code and the datasets used in our experiments are available in the supplementary material. In addition, proofs of the theorems are presented either in the main text or in Appendix C.

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

## A    ALGORITHM PSEUDOCODE

The SLCD pseudocode:

---
**Algorithm 1** Sparse Linear Causal Discovery ( SLCD) Algorithm.

---
**Inputs:**
  $\boldsymbol{X} \in \mathbb{R}^{n \times m}$, $N$, $M$, $\lambda$, $\sigma$, $\tau$
**for** $t = 1 : M$ **do**
    **Initialize:**
    $\boldsymbol{D}_0 \in \mathbb{R}^{n \times n} : randomly$
    **if** (t == 1) **then**
        $J_{min} \leftarrow J(\boldsymbol{D}_0)$
        $\boldsymbol{D}_{opt} \leftarrow \boldsymbol{D}_0$
    **end if**
    **for** $k = 1 : N$ **do**
        $\boldsymbol{D} \leftarrow$ Solve (12) (e.g. fmincon (MATLAB))
        **if** $J_{min} > J(\boldsymbol{D})$ **then**
            $J_{min} \leftarrow J(\boldsymbol{D})$
            $\boldsymbol{D}_{opt} \leftarrow \boldsymbol{D}$
        **end if**
    **end for**
**end for**
**return** $\boldsymbol{D}_{opt}$

---

## B    STRUCTURAL RECOVERY RESULTS OF SLCD ACROSS DIFFERENT DATASETS

Table 5 shows the recovered structural matrix using SLCD as well as the ground truth structural matrix for each dataset with $(\sigma, \lambda) = (0.3, 5)$.

## C    PROOFS

### C.1    PROOF OF THEOREM 2

Let $\boldsymbol{x} \in \mathbb{R}^n$ be a random vector with the following causal structure:

$$\boldsymbol{x} = \sum_{i=1}^{\infty} \boldsymbol{D}_i \boldsymbol{x}^i, \tag{20}$$

where $\boldsymbol{x}^i$ is the vector obtained by elementwise raising of $\boldsymbol{x}$ to the $i$-th power, and $\boldsymbol{D}_i$ is the corresponding coefficient matrix.

| Dataset | Structure matrix | Estimated structure matrix | $(\sigma, \lambda)$ |
|---|---|---|---|
| Dataset 1 | $\begin{bmatrix} 1 & 0 & 0 \\ 2 & 1 & 0 \\ 0.4 & 1 & 0 \end{bmatrix}$ | $\begin{bmatrix} 0 & 0.5 & 4.4\times10^{-7} \\ 2 & 1 & 1.1\times10^{-6} \\ -1.2\times10^{-7} & 0.2 & 0 \end{bmatrix}$ | $(0.3, 5)$ |
| Dataset 2 | $\begin{bmatrix} 1 & 0 & 0 & 0 \\ 0 & 1 & 0 & 0 \\ 0.3 & 1 & 0 & 0 \\ 1 & 2 & 0 & 0 \end{bmatrix}$ | $\begin{bmatrix} 1 & 0 & -8.8\times10^{-4} & 0 \\ 0 & 1 & -3.3\times10^{-4} & 0 \\ 0.3 & 0 & -2.7\times10^{-4} & 0 \\ 1.002 & 1.999 & 0 & 0 \end{bmatrix}$ | $(0.3, 5)$ |
| Dataset 3 | $\begin{bmatrix} 1 & 0 & 0 & 0 & 0 \\ 0 & 1 & 0 & 0 & 0 \\ 1 & 3 & 0 & 0 & 0 \\ 0 & 2 & 0 & 0 & 0 \\ 2 & 1 & 0 & 0 & 0 \end{bmatrix}$ | $\begin{bmatrix} 0.999 & 0.0497 & 0 & 0 & 0 \\ 0 & 1.000 & 0 & 0 & 0.0102 \\ 0.976 & 3.049 & 0 & 0 & 0 \\ -0.0147 & 1.999 & 0 & 0 & 0 \\ 1.990 & 1.099 & 0 & 0 & 0 \end{bmatrix}$ | $(0.3, 5)$ |
| Dataset 4 | $\begin{bmatrix} 1 & 0 & 0 & 0 & 0 & 0 \\ 0 & 1 & 0 & 0 & 0 & 0 \\ 0 & 0 & 1 & 0 & 0 & 0 \\ 1 & 0 & 0.3 & 0 & 0 & 0 \\ 2 & 3 & 0 & 0 & 0 & 0 \\ 0 & 2 & 0.5 & 0 & 0 & 0 \end{bmatrix}$ | $\begin{bmatrix} 0.999 & -0.009 & 0 & 0 & 0 & 0 \\ 0.016 & 0.999 & 0 & 0 & 0 & 0 \\ -.0432 & 0 & 0.997 & 0 & 0 & 0 \\ 0.987 & 0 & 0.3019 & 0 & 0 & 0 \\ 2.048 & 2.982 & 0 & 0 & 0 & 0 \\ 0 & 1.995 & 0.483 & 0 & 0 & 0 \end{bmatrix}$ | $(0.3, 5)$ |
| Dataset 5 | $\begin{bmatrix} 1 & 0 & 0 & 0 & 0 & 0 & 0 \\ 0 & 1 & 0 & 0 & 0 & 0 & 0 \\ 0 & 0 & 1 & 0 & 0 & 0 & 0 \\ 1 & 0 & 0.5 & 0 & 0 & 0 & 0 \\ 0 & 1 & 2 & 0 & 0 & 0 & 0 \\ 1 & 0 & 3 & 0 & 0 & 0 & 0 \\ 0 & 1 & 1 & 0 & 0 & 0 & 0 \end{bmatrix}$ | $\begin{bmatrix} 0.997 & .0525 & 0 & 0 & 0 & 0 \\ -0.082 & 0.994 & 0 & 0 & 0 & 0 & 0 \\ 0.057 & 0 & 0.998 & 0 & 0 & 0 & 0 \\ 1.025 & 0 & 0.491 & 0 & 0 & 0 & 0 \\ 0 & 0.956 & 2.024 & 0 & 0 & 0 & 0 \\ 1.168 & 0 & 2.986 & 0 & 0 & 0 & 0 \\ 0 & 0.975 & 1.025 & 0 & 0 & 0 & 0 \end{bmatrix}$ | $(0.3, 5)$ |

Table 5: True structural matrix and the output of SLCD.

To calculate the covariance matrix of the data, we need to compute $\mathbb{E}[x_i x_j]$. Using equation 20, we have

$$\mathbb{E}[x_i x_j] = \mathbb{E}\left[ \left( \sum_{l=1}^{\infty} \boldsymbol{d}_i^{(l)T} \boldsymbol{x}^l \right) \left( \sum_{k=1}^{\infty} \boldsymbol{d}_j^{(k)T} \boldsymbol{x}^k \right) \right], \tag{21}$$

where $\boldsymbol{d}_i^{(l)T}$ and $\boldsymbol{d}_j^{(k)T}$ are the $i^{\text{th}}$ and $j^{\text{th}}$ rows of the matrix $\boldsymbol{D}_l$ and $\boldsymbol{D}_k$, respectively.

Applying this procedure to all pairs of entries, we obtain

$$\boldsymbol{\Sigma} = \sum_{i=1}^{\infty} \sum_{j=1}^{\infty} \boldsymbol{D}_i \boldsymbol{\sigma}_{ij} \boldsymbol{D}_j^T, \tag{22}$$

where $\boldsymbol{\sigma}_{ij}$ is a diagonal matrix whose $l^{\text{th}}$ diagonal element is $\mathbb{E}[x_l^i x_l^j]$ for all $l \in \{1, 2, \ldots, n\}$.

## C.2 PROOF OF THEOREM 3

Let $F(\boldsymbol{D}), \boldsymbol{\Sigma}, \boldsymbol{\sigma}$, and $\boldsymbol{X}$ be as in Theorem 3. Suppose $\boldsymbol{D}^*$ satisfies both $F(\boldsymbol{D}) = \boldsymbol{0}$ and $\boldsymbol{D}^* \boldsymbol{X} = \boldsymbol{X}$. Let $\boldsymbol{D}$ be another solution of $F(\boldsymbol{D}) = \boldsymbol{0}$ satisfying $\boldsymbol{D}\boldsymbol{X} = \boldsymbol{X}$, and define $\boldsymbol{\Delta} = \boldsymbol{D} - \boldsymbol{D}^*$. Then $\boldsymbol{\Delta}\boldsymbol{X} = \boldsymbol{0}$. We expand

$$F(\boldsymbol{D}) = F(\boldsymbol{D}^* + \boldsymbol{\Delta}) = \boldsymbol{\Sigma} - (\boldsymbol{D}^* + \boldsymbol{\Delta})\boldsymbol{\sigma}(\boldsymbol{D}^* + \boldsymbol{\Delta})^T$$
$$= \boldsymbol{\Sigma} - \boldsymbol{D}^*\boldsymbol{\sigma}\boldsymbol{D}^{*T} - (\boldsymbol{D}^*\boldsymbol{\sigma}\boldsymbol{\Delta}^T + \boldsymbol{\Delta}\boldsymbol{\sigma}\boldsymbol{D}^{*T}) - \boldsymbol{\Delta}\boldsymbol{\sigma}\boldsymbol{\Delta}^T$$

Using $F(\boldsymbol{D}^*) = \boldsymbol{0}$ gives

$$F(\boldsymbol{D}) = -\left(\boldsymbol{D}^*\boldsymbol{\sigma}\boldsymbol{\Delta}^T + \boldsymbol{\Delta}\boldsymbol{\sigma}\boldsymbol{D}^{*T}\right) - \boldsymbol{\Delta}\boldsymbol{\sigma}\boldsymbol{\Delta}^T = -\mathcal{L}(\boldsymbol{\Delta}) - \mathcal{Q}(\boldsymbol{\Delta}),$$

where

$$\mathcal{L}(\boldsymbol{\Delta}) = \boldsymbol{D}^*\boldsymbol{\sigma}\boldsymbol{\Delta}^T + \boldsymbol{\Delta}\boldsymbol{\sigma}\boldsymbol{D}^{*T}, \qquad \mathcal{Q}(\boldsymbol{\Delta}) = \boldsymbol{\Delta}\boldsymbol{\sigma}\boldsymbol{\Delta}^T.$$

Since $F(\boldsymbol{D}) = \boldsymbol{0}$, we obtain

$$\mathcal{L}(\boldsymbol{\Delta}) + \mathcal{Q}(\boldsymbol{\Delta}) = \boldsymbol{0}, \qquad \boldsymbol{\Delta}\boldsymbol{X} = \boldsymbol{0}. \tag{23}$$

Define the subspace

$$\mathcal{S} = \{\, \boldsymbol{\Delta} \in \mathbb{R}^{n \times n} \mid \boldsymbol{\Delta}\boldsymbol{X} = \boldsymbol{0} \,\}.$$

By assumption, no nonzero $\boldsymbol{\Delta} \in \mathcal{S}$ makes $\boldsymbol{D}^* \boldsymbol{\sigma} \boldsymbol{\Delta}^T$ skew-symmetric. This implies that

$$\mathcal{L}(\boldsymbol{\Delta}) = \boldsymbol{0}, \, \boldsymbol{\Delta} \in \mathcal{S} \quad \Longrightarrow \quad \boldsymbol{\Delta} = \boldsymbol{0}.$$

Thus $\mathcal{L}$ is injective on $\mathcal{S}$, and consequently there exists $c > 0$ such that

$$\|\mathcal{L}(\boldsymbol{\Delta})\|_F \geq c \, \|\boldsymbol{\Delta}\|_F, \qquad \forall \boldsymbol{\Delta} \in \mathcal{S}. \tag{24}$$

For the quadratic term we use submultiplicativity:

$$\|\mathcal{Q}(\boldsymbol{\Delta})\|_F = \|\boldsymbol{\Delta}\boldsymbol{\sigma}\boldsymbol{\Delta}^T\|_F \leq \|\boldsymbol{\sigma}\|_2 \, \|\boldsymbol{\Delta}\|_F^2, \qquad \forall \boldsymbol{\Delta} \in \mathcal{S}. \tag{25}$$

Combining (23), (24), and 25 yields

$$c \, \|\boldsymbol{\Delta}\|_F \leq \|\mathcal{L}(\boldsymbol{\Delta})\|_F = \|\mathcal{Q}(\boldsymbol{\Delta})\|_F \leq \|\boldsymbol{\sigma}\|_2 \, \|\boldsymbol{\Delta}\|_F^2,$$

and hence

$$\|\boldsymbol{\Delta}\|_F \geq \frac{c}{\|\boldsymbol{\sigma}\|_2}.$$

This establishes that no other solution lies within the ball $\{\, \boldsymbol{D} : \|\boldsymbol{D} - \boldsymbol{D}^*\|_F < r \,\}$ of radius $r = \frac{c}{\|\boldsymbol{\sigma}\|_2}$ around $\boldsymbol{D}^*$. Hence $\boldsymbol{D}^*$ is isolated and locally unique.

### C.3 PROOF OF THEOREM 4

*Proof.* Recall that $F(\boldsymbol{D}) = \boldsymbol{\Sigma} - \boldsymbol{D}\boldsymbol{\sigma}\boldsymbol{D}^\top$ and that $\boldsymbol{D}^*$ satisfies $F(\boldsymbol{D}^*) = \boldsymbol{0}$ and $\boldsymbol{D}^*\boldsymbol{X} = \boldsymbol{X}$. Let the covariance matrix be perturbed as $\boldsymbol{\Sigma} \mapsto \boldsymbol{\Sigma} + \Delta\boldsymbol{\Sigma}$, and let $\boldsymbol{D}$ satisfy the perturbed relation

$$\boldsymbol{\Sigma} + \Delta\boldsymbol{\Sigma} - \boldsymbol{D}\boldsymbol{\sigma}\boldsymbol{D}^\top = \boldsymbol{0}, \qquad \boldsymbol{D}\boldsymbol{X} = \boldsymbol{X}.$$

Define $\boldsymbol{\Delta} = \boldsymbol{D} - \boldsymbol{D}^*$ and

$$\mathcal{S} = \{\, \boldsymbol{\Delta} \in \mathbb{R}^{n \times n} \mid \boldsymbol{\Delta}\boldsymbol{X} = \boldsymbol{0} \,\}.$$

Then $\boldsymbol{\Delta} \in \mathcal{S}$, and expanding $F(\boldsymbol{D})$ gives

$$\boldsymbol{0} = \boldsymbol{\Sigma} + \Delta\boldsymbol{\Sigma} - \boldsymbol{D}\boldsymbol{\sigma}\boldsymbol{D}^\top = \boldsymbol{\Sigma} + \Delta\boldsymbol{\Sigma} - (\boldsymbol{D}^* + \boldsymbol{\Delta})\boldsymbol{\sigma}(\boldsymbol{D}^* + \boldsymbol{\Delta})^\top$$
$$= \underbrace{\boldsymbol{\Sigma} - \boldsymbol{D}^*\boldsymbol{\sigma}\boldsymbol{D}^{*\top}}_{\boldsymbol{0}} - \underbrace{(\boldsymbol{D}^*\boldsymbol{\sigma}\boldsymbol{\Delta}^\top + \boldsymbol{\Delta}\boldsymbol{\sigma}\boldsymbol{D}^{*\top})}_{\mathcal{L}(\boldsymbol{\Delta})} - \underbrace{\boldsymbol{\Delta}\boldsymbol{\sigma}\boldsymbol{\Delta}^\top}_{\mathcal{Q}(\boldsymbol{\Delta})} + \Delta\boldsymbol{\Sigma}.$$

Hence

$$\mathcal{L}(\boldsymbol{\Delta}) + \mathcal{Q}(\boldsymbol{\Delta}) = \Delta\boldsymbol{\Sigma}, \qquad \boldsymbol{\Delta}\boldsymbol{X} = \boldsymbol{0}. \tag{26}$$

As in the proof of Theorem 3, the assumption that no nonzero $\boldsymbol{\Delta} \in \mathcal{S}$ makes $\boldsymbol{D}^*\boldsymbol{\sigma}\boldsymbol{\Delta}^\top$ skew-symmetric implies that $\mathcal{L}$ is injective on $\mathcal{S}$. Consequently, there exists $c > 0$ such that

$$\|\mathcal{L}(\boldsymbol{\Delta})\|_F \geq c \, \|\boldsymbol{\Delta}\|_F, \qquad \forall \boldsymbol{\Delta} \in \mathcal{S}, \tag{27}$$

where

$$c = \inf_{\substack{\boldsymbol{\Delta} \in \mathcal{S} \\ \|\boldsymbol{\Delta}\|_F = 1}} \|\mathcal{L}(\boldsymbol{\Delta})\|_F.$$

For the quadratic term, submultiplicativity yields

$$\|\mathcal{Q}(\boldsymbol{\Delta})\|_F = \|\boldsymbol{\Delta}\boldsymbol{\sigma}\boldsymbol{\Delta}^\top\|_F \leq \|\boldsymbol{\sigma}\|_2 \, \|\boldsymbol{\Delta}\|_F^2, \qquad \forall \boldsymbol{\Delta} \in \mathcal{S}. \tag{28}$$

Combining (26) and 28 gives

$$c \, \|\boldsymbol{\Delta}\|_F \leq \|\mathcal{L}(\boldsymbol{\Delta})\|_F = \|\Delta\boldsymbol{\Sigma} - \mathcal{Q}(\boldsymbol{\Delta})\|_F \leq \|\Delta\boldsymbol{\Sigma}\|_F + \|\mathcal{Q}(\boldsymbol{\Delta})\|_F \leq \|\Delta\boldsymbol{\Sigma}\|_F + \|\boldsymbol{\sigma}\|_2 \, \|\boldsymbol{\Delta}\|_F^2. \tag{29}$$

Let $r = \|\boldsymbol{\Delta}\|_F$. Then from 29, $r$ satisfies the quadratic inequality

$$\|\boldsymbol{\sigma}\|_2 \, r^2 - c \, r + \|\Delta\boldsymbol{\Sigma}\|_F \geq 0. \tag{30}$$

If the discriminant $c^2 - 4\|\boldsymbol{\sigma}\|_2 \|\Delta\boldsymbol{\Sigma}\|_F$ is negative, the inequality imposes no constraint on $r$. If

$$\|\Delta\boldsymbol{\Sigma}\|_F \; < \; \frac{c^2}{4\,\|\boldsymbol{\sigma}\|_2},$$

then (30) implies that $r$ cannot lie strictly between the two real roots. That is, $\|\boldsymbol{D} - \boldsymbol{D}^*\|_F = r$ cannot satisfy

$$\frac{c - \sqrt{c^2 - 4\|\boldsymbol{\sigma}\|_2 \|\Delta\boldsymbol{\Sigma}\|_F}}{2\,\|\boldsymbol{\sigma}\|_2} \; < \; r \; < \; \frac{c + \sqrt{c^2 - 4\|\boldsymbol{\sigma}\|_2 \|\Delta\boldsymbol{\Sigma}\|_F}}{2\,\|\boldsymbol{\sigma}\|_2}.$$

$\square$

