# OpenReview forum: "INDUCED COVARIANCE FOR CAUSAL DISCOVERY IN LINEAR SPARSE STRUCTURES"
_ICLR.cc/2026/Conference — Submitted to ICLR 2026_

### Official Review · Reviewer_TCJT · 2025-10-16

**Soundness:** 1
**Presentation:** 1
**Contribution:** 1
**Rating:** 2
**Confidence:** 4

**Summary:**

This paper proposes a method called Sparse Linear Causal Discovery (SLCD), which aims to identify causal relationships under the assumption of a sparse linear structure among variables, to help causal discovery in the low data sample size regime.

The authors claim that the method avoids conditional independence tests or score search. Instead, they optimize over a structural matrix D that reconstructs the data and satisfies a so-called induced covariance constraint. They then extend this idea to nonlinear settings via a Taylor expansion argument and provide several theoretical propositions about local uniqueness and sensitivity.

Experimental comparisons with existing methods on synthetic data reportedly show improvements.

**Strengths:**

1. The topic of causal discovery in the limited data regime is a relevant and active area, and the motivation of avoiding CI tests under limited data is valid.

**Weaknesses:**

1. **The theoretical framework is poorly grounded:  trivial setting, self-contradictory results, no identifiability discussion at all.**
 - The authors interpret causal relations as deterministic linear transformations (Equations 2,3,4).
 - This makes the contribution trivial.  Let the covariance matrix among observed variables be cov(X).  The constraint D*cov(X)*D^T = cov(X) is just to require D to be orthogonal against cov(X).  There is no structural / causal implications for it.
 - Hence, the claimed contribution "introduce the concept of induced covariance, a statistical property implied by causal structures" is just following directly from linear algebra and does not constitute a meaningful causal result.
 - Such fully deterministic linear transformation is also contradicted to authors' own technical progression. E.g., at Equation 1 the hidden noise terms "u_i" are presented.
 - There is no identifiability analysis at all: no argument that the true causal structure can be recovered under any set of assumptions. Theorems 3 and 4 are stated abstractly without clear causal interpretation or practical significance.
 - The authors may also have misunderstandings about identifiability notion.  E.g., at line 151, "This is a common problem in causal discovery as multiple graphs can describe the same data Spirtes et al. (2001)."  It is actually not.  The problem that authors are presenting is simply rewriting the linear transformation.

2. **Technical development is messy and difficult to follow: sloppy and inconsistent exposition**
 - For example, in the problem statement (line 110), "We define I as the set of indices for independent variables and D as the set of indices for dependent variables. "  However, the meanings for "independent variables" and "dependent variables" are not defined. The notation "\mathcal{I}" is never used either.
 - Line 118, "a vector with fewer than τ non-zero elements" -- I believe the authors intended to say "no more than".
 - As mentioned earlier, in Equation 3, the authors define the causal model without exogenous noise, yet noise was present in Equation 1 when introducing the SCM.

3. **Unprofessional and low-quality writing:**
 - The narrative flow is chaotic: basic definitions appear late, theorems are introduced without intuition, etc.
 - Sentences like "by an average of 35% in precision and 41.5% in recall across all tested datasets." appear in abstract, without any explanation to the experimental setting.
 - The "proofs" and "theorems" are written more like algebraic manipulations than proper statements with clear assumptions and claims.

**Questions:**

n/a

---

> ### Author Response · Authors · 2025-11-28
>
> Thank you for your feedback.
>
> **W1**:
>
> **(bullets 1–4)**
> Our formulation considers a setting in which a subset of variables (“dependent variables”) is generated by an unknown sparse linear mechanism applied to another subset (“independent variables”).
> The noiseless deterministic structural relations that we consider in this work, i.e., SCMs without exogenous noise, arise naturally in many domains, including physics, biology, and systems governed by differential equations. For instance:
>
> [1] Li, Loka, et al. "On causal discovery in the presence of deterministic relations." Advances in Neural Information Processing Systems 37 (2024): 130920-130952.).
>
> [2] Yang, Yuqin, et al. "Causal discovery in linear structural causal models with deterministic relations." Conference on Causal Learning and Reasoning. PMLR, 2022.
>
> The induced covariance constraint is not an arbitrary orthogonality condition but a structural consequence of the SCM form we consider. While the constraint is derived using linear-algebraic tools, it follows directly from the assumed causal mechanism and is used to restrict the feasible set of solutions. In other words, the induced covariance is a direct implication of the causal relations. This dependency can be seen as its functional dependence on $\boldsymbol{D}$.
> The presentation of exogenous noise in Eq. (1) was meant to provide general SCM context. Our model corresponds to the deterministic/noiseless special case $(u_i = 0)$, which as mentioned above, arise naturally in many domains such as physics, biology, and systems governed by differential equations. We will clarify this more clearly in the revised manuscript.
>
>
> **(bullet 5)**
> Global identifiability guarantees require additional assumptions, such as access to interventional data, that are not available in our setting. Our goal is different: we study what information can be recovered purely from observational data under a deterministic sparse structure. Accordingly, Theorem 3 provides a local uniqueness analysis and demonstrates a region of attraction around the true solution within which the true structure can be uniquely determined. Theorem 4 provides a sensitivity analysis, showcasing how the solution reacts to imprecision in the covariance matrix.
>
> **(bullet 6)**
> Regarding the comment about Spirtes et al. (2001), our statement referred to the broader challenge that observational distributions may correspond to multiple structural graphs. We have rewritten this sentence to avoid confusion.
>
> **W2**:
> We appreciate these observations and have revised the manuscript extensively to improve clarity.
> We now explicitly define “independent variables,” “dependent variables,” and the index set $\mathcal{D}$ when they are first introduced.
>
> We corrected the phrasing on line 118 to “no more than τ non-zero elements” and performed an additional proofreading pass to eliminate similar mistakes.
>
> As noted above, we revised the section around Equation (3) to clarify the considered model.
>
> **W3**:
> We acknowledge the reviewer’s concerns and have substantially revised the manuscript to improve readability and flow. This includes:
>
> * clarifying the experimental setup in the abstract
> * rewriting the proofs to follow a standard theorem–assumption–claim structure

---

### Official Review · Reviewer_mcfV · 2025-10-27

**Soundness:** 2
**Presentation:** 3
**Contribution:** 2
**Rating:** 4
**Confidence:** 3

**Summary:**

This paper proposes a method for causal discovery in linear SEMs based on a novel constraint termed induced covariance, where the observed covariance matrix $\Sigma$ must factorize as $D \sigma D^T$ with $D$ being the structural matrix and $\sigma$ a diagonal noise variance matrix. The authors frame causal discovery as a matrix factorization problem with reconstruction and covariance constraints and optimize for a sparse D. This paper claims identifiability under sparsity and independence assumptions and propose a variational approach (SLCD) to recover the causal graph from empirical covariance. Experiments on synthetic data show improved performance over standard causal discovery algorithms.

**Strengths:**

a. Introduces a new formulation for structure learning that leverages second-order constraints, avoiding conditional independence tests.

b. The proposed optimization objective is intuitive, combining covariance factorization and data reconstruction with sparsity-promoting penalties.

c. Theoretical results show local uniqueness of the true structural matrix under certain conditions.

d. Experiments on simulated data suggest improved recovery of sparse DAGs in low-sample regimes.

**Weaknesses:**

a. Theorem 1, which establishes the covariance factorization $\Sigma = D \sigma D^T$, is not new and was previously formalized in, e.g., Sullivant et al. (2010) through trek separation theory. The paper should clearly cite this foundational work.

b. The main theoretical guarantee shows only local uniqueness, not global identifiability. This means alternative structures could still satisfy the constraints elsewhere in the parameter space, so identifiability is not ensured in the full sense.

c. The experimental evaluation is limited to synthetic datasets. Real-world evaluations or tests under assumption violations (e.g., with latent confounders) would improve the empirical case.

Sullivant S, Talaska K, Draisma J. Trek separation for Gaussian graphical models[J]. 2010.

**Questions:**

a. Could you clarify how your method would behave when the diagonal noise assumption is violated (e.g., with latent confounders)?

b. Could you provide examples where local uniqueness fails?

c. Is the nonlinear extension implemented, or is it purely theoretical at this stage?

---

> ### Author Response · Authors · 2025-11-28
>
> Thank you for your feedback.
>
> **Weaknesses**:
>
> **a)** We were not aware of the mentioned work, and we will add the citation in the revised version. However, our setting differs from that work in an important way: Sullivant et al. assume that the variables in the system are jointly Gaussian and base their analysis on this assumption. In contrast, in our work the independent variables (causal variables) may follow any arbitrary distribution, and the distribution of the dependent variables (effect variables) is determined both by the distribution of the independent variables and the linear relations linking them. Our framework imposes no requirement that the variables be jointly Gaussian.
>
> **b)** Global identifiability guarantees require additional assumptions, such as access to interventional data, that are not available in our setting. Our goal is different: we study what information can be recovered purely from observational data under a deterministic sparse structure. Accordingly, Theorem 3 provides a local uniqueness analysis and demonstrates a region of attraction around the true solution within which the true structure can be uniquely determined. Theorem 4 provides a sensitivity analysis, showcasing how the solution reacts to imprecision in the covariance matrix.
>
>
> **c)** We will add simulations on the real-world Sachs dataset in the revised version.
>
> **Questions**:
>
> **a)** This is not part of our model assumptions in this work. Here it is assumed that there are no confounders in the dataset and that all variables are observed.
>
> **b)** Local uniqueness can fail when the dataset does not have enough complexity. By complexity, we mean that if, for instance, one causal variable is responsible for generating all other variables, then the dataset is not complex enough. This is shown in our simulations, as the method performs poorly when the dataset has only three variables, with only one being causal. The method does not perform well in that situation and local uniqueness fails.
>
> **c)** The nonlinear extension is theoretical at this stage; however, we will be adding simulations on a real-world dataset.

---

### Official Review · Reviewer_sHRJ · 2025-10-30

**Soundness:** 1
**Presentation:** 1
**Contribution:** 1
**Rating:** 2
**Confidence:** 4

**Summary:**

This paper introduces a causal discovery method called Sparse Linear Causal Discovery (SLCD) for the scenarios where variables exhibit linearly sparse relationships. SLCD leverages the structural matrix's ability to reconstruct data and the statistical properties it imposes on the data to identify the correct structural matrix. SLCD does not rely on independence tests or graph fitting procedures.

**Strengths:**

Unfortunately, I find no obvious strength of this paper.

**Weaknesses:**

1. The theoretical results in this paper cannot demonstrate the superiority of their proposed SLCD.

- Both Theorem 3 and Theorem 4 assume there is a solution $D$ s.t. $F(D) = 0$ and $DX = X$. However, such a $D$ does not exists if there are noise terms, which is a standard setting in the previous literature on causality.

- Even there exists such a $D$, Theorem 3 and Theorem 4 only demonstrate that two solutions $D, D'$ s.t. $F(D) = F(D') = 0$ and $D X = D' X = X$ are close to each other, which does not necessarily imply that $D$ is close to the ground truth.

2. The experimental results are not convincing.

- As claimed in Section 5, SLCD can be extended to scenarios in which the SCMs governing the causal relations are nonlinear. Why not conduct experiments on nonlinear SCMs?

- Why not conduct experiments on real-world datasets?

3. The presentation of this paper is not friendly to readers in the causality community. For instance, in Section 3, this paper introduce the high-level insights in the setting without noise terms, which does not align with most previous literature on causality.

**Questions:**

1. In the second graph of Introduction, the authors claim that causal discovery methods are generally classified into two categories: constraint-based methods and score-based methods. However, according to my experience, researchers in the causality community used to divide causal discovery methods into three categories: constraint-based, score-based, and functional causal model-based methods. LiNGAM is typically regarded as a FCM-based method rather than a constraint-based method.

2. The authors classify variables of interest into independent variables and dependent variables, but they don't provide formal definitions of independent/dependent variables. According to my understanding, it seems that independent variables are root variables while dependent variables are non-root variables. But the authors also claim that each dependent variable is a function of a subset of independent variables that are considered as its parents, why cannot a dependent variable have another dependent variable as its parent?

3. There is a typo in Equation (10), where $d_i^T \sigma d_j^T$ should be $d_i^T \sigma d_j$.

4. In Equation (11), why do you minimize the rank and the trace of $D$? In other words, why do you think the ground truth is the one with the minimal rank and trace?

5. Also in Equation (11), why do you impose the constraint that $\forall i, ||d_ i^T||_ 0 = \tau$ rather than $\forall i, ||d_ i^T||_0 \leq \tau$?

6. Where is footnote 1 in Table 1?

---

> ### Author Response · Authors · 2025-11-28
>
> Thank you for your feedback.
>
> **W1**:
> We consider the setting where the causal relations among the variables are linear and noiseless (deterministic) but unknown, and the goal is to recover this unknown relation between the causal variables (independent variables) and the effects (the dependent variables). In this setting, the objective is to recover the deterministic linear causal structure using only the given dataset, without access to interventional data or other external information.
> Please note that Global identifiability guarantees require additional assumptions, such as access to interventional data, that are not available in our setting. Our goal is different: we study what information can be recovered purely from observational data under a deterministic sparse structure.
>
> **Purpose of Theorem 3 and Theorem 4**
> Theorem 3 and Theorem 4 are intended to provide insights into the behavior of the optimization problem. More specifically Theorem 3 provides a local uniqueness analysis and demonstrates a region of attraction around the true solution within which the true structure can be uniquely determined. Theorem 4 provides a sensitivity analysis, showcasing how the solution reacts to imprecision in the covariance matrix. In other words
> * Based on the optimization problem, the true $\boldsymbol{D}$ is a solution of the optimization problem.
> * Theorem 3 studies the behavior around the true $\boldsymbol{D}$: it shows that this solution is isolated and that there exists a radius around the true $\boldsymbol{D}$ within which the solution can be uniquely determined.
> * Theorem 4 provides a perturbation analysis of the solution, showing what happens to the true $\boldsymbol{D}$ when the covariance matrix is perturbed.
>
> **W2**:
> We will update the revised version to include simulation results on a nonlinear real-world Sachs dataset.
>
> **W3**:
> The deterministic setting for causality without exogenous noise is allowed under the SCM framework and can address a variety of problems, including those in physics, biology, and systems modeled through differential equations, where the effects can be deterministically known from the values of the cause variables. For instance:
>
> [1] Li, Loka, et al. "On causal discovery in the presence of deterministic relations." Advances in Neural Information Processing Systems 37 (2024): 130920-130952.).
>
> [2] Yang, Yuqin, et al. "Causal discovery in linear structural causal models with deterministic relations." Conference on Causal Learning and Reasoning. PMLR, 2022.
>
> **Q1**:
> We used the same categorization adopted in the following work:
>
> Bernhard Schölkopf and Julius von Kügelgen. From statistical to causal learning. ICM, 2022.
>
> **Q2**:
> We will provide a more formal and clear definition of independent and dependent variables in the revised version. We thank the reviewer for the comment.
> The reviewer’s understanding is correct: independent variables correspond to root variables, and dependent variables correspond to non-root variables. Due to linearity and deterministic relations, any dependency chain always traces back to root variables. For example:
>
>  Let $x_1$be a root variable and $x_2=2x_1$.
>  Let $x_3=3x_2$.
>  Then $x_3=3\times(2x_1)=6x_1$, which always returns to the root variable.
>
> **Q3**:
> We agree with the reviewer and thank them for pointing this out. It will be corrected in the revised version.
>
> **Q4**:
> In our modeling, the structure is assumed to be sparse, meaning only a small number of causal variables (root variables) exist in the system. The number of these variables determines the rank. Since the number of causal variables is assumed to be less than the total number of variables (sparsity condition), the solution with lower rank is aligned with the true structure. The trace term helps eliminate the trivial solution where all variables are considered independent. These two constraints align with the structural properties of the true $\boldsymbol{D}$, which is why they are used.
>
> **Q5**:
> We thank the reviewer for the comment. This is a typo and will be revised throughout the manuscript.
>
> **Q6**:
> We thank the reviewer for the comment. This will be revised in the updated version.

---

### Official Review · Reviewer_cq5J · 2025-10-31

**Soundness:** 3
**Presentation:** 2
**Contribution:** 3
**Rating:** 4
**Confidence:** 3

**Summary:**

This paper discusses a method for accurately estimating causal structures even in scenarios with limited sample sizes. Conventional approaches that rely on independence tests or graph score optimization often suffer a significant loss of reliability when data are scarce. To address this issue, the authors propose a novel causal discovery algorithm called Sparse Linear Causal Discovery (SLCD), which leverages the structural matrix's induced covariance and reconstruction properties to recover the true causal structure. The method is particularly effective when the relationships among variables are linear and sparse. Furthermore, the paper extends the framework to handle nonlinear causal relations by using a Taylor-series-based polynomial approximation, enabling similar causal estimation under nonlinear transformations. The authors also analyze the local uniqueness and perturbation stability of the proposed method to provide theoretical insights into its identifiability properties. Experimental evaluations on simulated datasets demonstrate that SLCD outperforms existing approaches such as PC, GES, BIC exact search, and LiNGAM, achieving higher accuracy in recovering causal structures.

**Strengths:**

The paper introduces a novel approach to causal structure estimation based on the concept of induced covariance, offering a fresh perspective distinct from traditional statistical causal discovery methods. It demonstrates that, under the assumption of sparsity in the causal structure, the proposed method can accurately recover causal graphs even with a limited number of samples. This contributes to expanding the applicability of causal discovery to real-world scenarios where sample sizes are often small.

The authors also provide theoretical analyses of local uniqueness and perturbation stability, which lend a degree of theoretical soundness and reliability to the proposed method.

Through simulation-based experiments, the paper shows that the proposed algorithm achieves higher accuracy than many existing approaches, particularly in small-sample settings.

**Weaknesses:**

The paper lacks sufficient explanation of several critical assumptions, which raises major concerns about the soundness of its theoretical development. In particular, the derivation from Equation (7) to Equation (10) assumes that the variables $x_i$ are largely uncorrelated implicitly; without this assumption, the equations do not hold as presented. While such an assumption would make the derivation understandable, the paper provides almost no discussion of it. As a result, readers may struggle to follow the theoretical logic and, more importantly, may fail to recognize the key assumptions underlying the proposed method, potentially leading to misunderstandings about the scope and applicability of the technique.

Another significant limitation is that the evaluation is conducted only on simulated data. Although the simulation results appear ideal under the assumed conditions, the paper focuses on scenarios with limited sample sizes, a setting that often arises in real-world applications. Therefore, it is essential to demonstrate the method’s performance on real-world datasets with small sample sizes to validate that the assumed conditions are realistic and practically relevant.

Finally, the paper contains several inconsistencies and inaccuracies in its mathematical notation and presentation. For example, in Equation (1), the variables are denoted as $y$, whereas all subsequent equations use $x$, which is inconsistent. Moreover, in Equation (10), the right-hand side should correctly be $d_i^T \sigma d_j$. Such issues in the notation of key equations are numerous and suggest a lack of careful proofreading. Overall, these presentation flaws prevent the paper from meeting the standard of clarity and rigor expected for a top-tier conference submission.

**Questions:**

1. Could you clearly explain what assumptions are made in the theoretical derivation from Equation (7) to Equation (10)?

2. If available, could you include experimental results using real-world datasets?

---

> ### Author Response · Authors · 2025-11-28
>
> Thank you for your feedback.
>
> **Weaknesses**:
>
> The assumptions for the derivation from Equation (7) to Equation (10) are as follows. The model we consider for this work is an unknown linear deterministic relation that exists among the variables. The assumption is that all the variables in the problem are categorized into one of these categories: either they are causal variables (independent variables, which are mutually independent and therefore uncorrelated) or effect variables (dependent variables). It is assumed that the effect variables are related to the causal variables via a linear function. Equation (7) shows that linear combination in vector form for a variable $x_i$. The same goes for $x_j$. Equation (10) then follows from the multiplication and carrying the expectation.
>
> We will add simulations on the real-world Sachs dataset in the revised version.
>
> We thank the reviewer for raising the concerns. The use of different symbols for the causal variable formulation on our side was intended to separate the preliminary material from the original formulation of the problem in the next section. We have defined them completely and independently whenever we used them. We agree with the reviewer on Equation (10); this is a typo and will be fixed in the revised version. We thank the reviewer for raising that. We will also revise the whole paper to increase clarity and readability.
>
> **Questions**
>
> 1- As described above, the assumptions are the linearity and deterministic form of the variables being related to each other and each dependent variable being a linear combination of the independent variables.
>
> 2- We will be presenting the simulation result on the real-world Sachs dataset in the revised version.

---

### Meta-Review · Area_Chair_Xq6A · 2026-01-07

**Summary:**

This paper introduces a causal discovery method called Sparse Linear Causal Discovery (SLCD) for the scenarios where variables exhibit linearly sparse relationships. The authors claim that the method avoids conditional independence tests or score search. Instead, they optimize over a structural matrix that reconstructs the data and satisfies a so-called induced covariance constraint. They then extend this idea to nonlinear settings via a Taylor expansion argument and provide several theoretical propositions about local uniqueness and sensitivity.

While the idea is interesting, the reviewers raise multiple concerns: lacking sufficient explanation of several critical assumptions, limited evaluation (largely on simulated dataset, limited sample sizes), containing several inconsistencies and inaccuracies in its mathematical notation and presentation. More critical reviewers further find this paper with significant clarity issues. There is a strong consensus here that the paper is not well-written (with presentation scores ranging between 1-2).

The rebuttal on the other hand is largely on editorial clarification which does not provide additional empirical and theoretical insights to address the other concerns on assumptions and limited experiments. Given this, I do not believe the reviewers will change their perspective of this paper. The final scores in my opinion will align with the original scores which lean strongly towards rejection.

**Reviewer Concerns:**

The reviewers raise multiple concerns: lacking sufficient explanation of several critical assumptions, limited evaluation (largely on simulated dataset, limited sample sizes), containing several inconsistencies and inaccuracies in its mathematical notation and presentation. More critical reviewers further find this paper with significant clarity issues. There is a strong consensus here that the paper is not well-written (with presentation scores ranging between 1-2).

**Reviewer Scores:**

The original scores are 4242 with multiple critical concerns that were not addressed by the rebuttal. As such, I have no reason to believe that the reviewers would upgrade their scores.

---

### Decision · Program_Chairs · 2026-01-26

Reject